# Goat Breeding: A Possible Sustainable Way to Manage Xerophytic Thickets in Southwestern Madagascar

Josoa R. Randriamalala [1,*] and Dominique Hervé [2]

1   Ecole Supérieure des Sciences Agronomiques, Université d'Antananarivo, BP 175, Antananarivo 101, Madagascar

2   Institut de Recherche pour le Développement, SENS (IRD, CIRAD, Université Paul Valéry), 911 Avenue Agropolis, 34394 Montpellier, France; dominique.herve@ird.fr

*   Correspondence: rramarolanonana@yahoo.fr; Tel.: +261-(0)-33-14-441-32

**Abstract:** Spiny thickets or xerophytic thickets (XTs) are a type of shrubby vegetation found in the far south and southwest of Madagascar, the driest parts of the island. This type of vegetation, which is rich in endemic animal and plant species, is endangered. Extensive local goat breeding (*Capra hircus*, for meat and milk production) based on XT browsing is an important source of household income. The aim of this paper is to analyse the possibility of using improved goat breeding as an alternative to wood charcoal (WC) production and slash-and-burn agriculture (SBA), which are unsustainable activities. The literature on (i) the impacts of SBA, WC production, and goat browsing on the XT ecosystem and (ii) the income provided by these three activities is reviewed to determine the sustainability level of improved goat husbandry. SBA and WC production reduced XT biomass and XT cover area, while goat browsing alone, at a stocking rate of one head per hectare, did not affect XT leaf biomass production and shrub regeneration. Furthermore, batch breeding and flushing techniques provided the highest annual income, mainly from the sale of surplus animal products. This improved goat husbandry may be a sustainable alternative to WC production. However, increasing the current stocking rate is necessary to surpass the combined income of WC production and SBA. An estimation of the XT carrying capacity would offer a basis to assess whether this ecosystem would support a higher stocking rate.

**Keywords:** arid; deforestation; goat husbandry; Madagascar; rangeland; sustainability; slash-and-burn-agriculture; wood charcoal

## 1. Introduction

Spiny thickets or xerophytic thickets (XTs) are a type of natural shrubby vegetation dominated by *Didiereaceae* and *Euphorbiaceae* [1] that occupies the littoral part of the southwest and the far south of Madagascar (Figure 1). Xerophytic thickets are a type of vegetation with a high endemism rate [2] and provide a multifunctional space and such goods and services as arable land [3,4], food and medicinal plants [5,6], goat rangeland [7–9], and timber and fuelwood (wood charcoal and firewood [6,10,11]). Similar to dry forests, XTs are currently undergoing significant deforestation (annual forest loss > 1%; [3,4]). Slash-and-burn agriculture (SBA) is the main cause of this deforestation [3,12], while wood charcoal (WC) production and to a lesser extent goat browsing are the main causes of XT degradation [11,13,14]. A search for sustainable alternatives to these unsustainable practices is necessary to stem this XT degradation/deforestation.

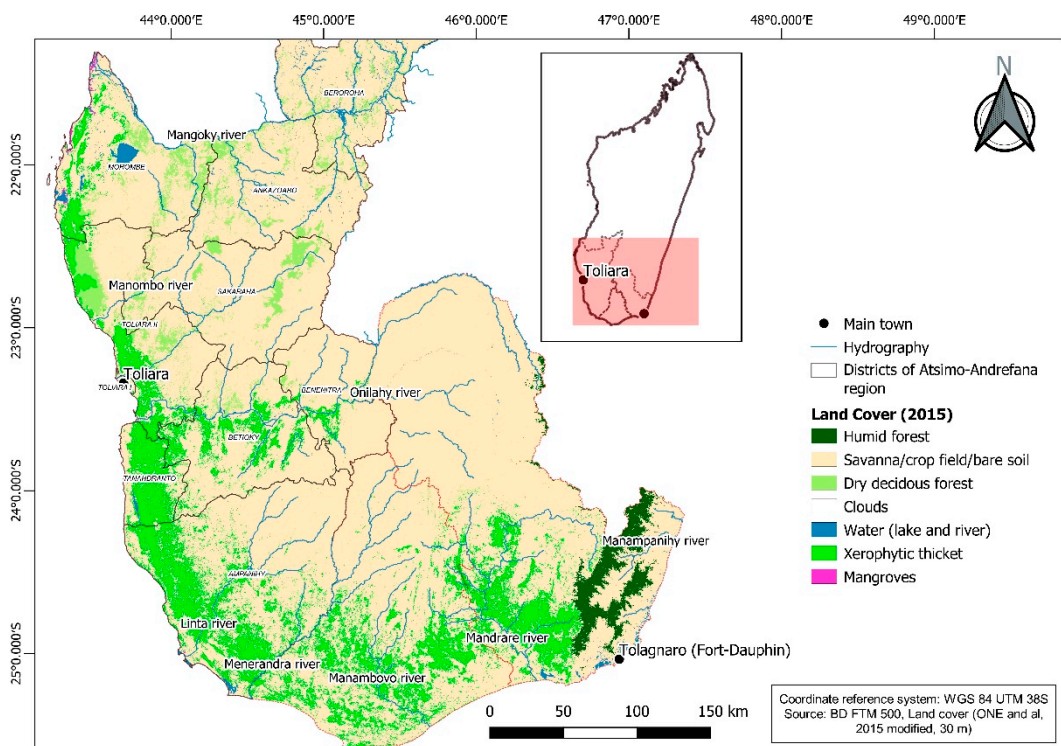

**Figure 1.** Southwestern Madagascar (the Atsimo-Andrefana region) and xerophytic thicket locations.

The development of rainfed agriculture would not be sufficient to reduce the dependence of households on forest resources, as the semi-arid character of the climate in southwestern Madagascar limits agricultural yields [15], while extensive local goat breeding (*Capra hircus*) is less impacted by rainfall variability [16,17]. As in other semi-arid areas, this extensive goat breeding has increased the livelihood resilience of populations living in and around the XTs to these climate risks [17]. However, goat breeding has a poor reputation in environmental terms. At the international level, many research papers show that intensive goat browsing reduces vegetation cover [18–22] and inhibits the regeneration of fodder species [23,24]. Goat browsing can also reduce biomass production and plant diversity [21,25,26]. However, when the stocking density is moderate, goat browsing can stimulate shrub shoot growth [27] and twig biomass production [26]. Furthermore, under moderate stocking conditions, goats can enhance seed dispersal in semi-arid ecosystems [28,29]. In addition, alternative forms of pasture management, such as the incorporation of planned rest periods (the opposite of continuous browsing), may reduce the negative effects of browsing on rangeland and increase plant cover [30].

This paper analyses, by a regional meta-review of the literature, the possibility of promoting goat breeding practices that could have a low environmental impact on the natural ecosystem and be economically viable for the livelihoods of local populations as an alternative to unsustainable activities such as WC production and SBA, which degrade and reduce the XT cover in southwestern Madagascar. The status of Malagasy XTs and their goat rangeland function are first addressed. Then, the impacts of the main activities of local populations (SBA, WC production, and goat browsing) on XTs are analysed. Next, the current goat breeding practices and their possible improvement are discussed. Finally, recommendations on reconciling economically viable goat breeding with sustainable XT management are proposed and discussed.

## 2. Methods

A bibliographical survey on Malagasy XTs was conducted for the purposes of this study. The research areas concerned (i) Malagasy XTs, (ii) the deforestation of Malagasy XTs, (iii) the degradation of Malagasy XTs, and (iv) goat breeding in and around XT

vegetation. References related to these topics were obtained using Google and Google Scholar search engines. The aim of this bibliographical survey was to simultaneously review the current state of knowledge on (i) the functioning of the XT ecosystem (in terms of diversity, structure, and natural regeneration), (ii) the effects of human practices on the XT ecosystem (SBA, WC production, and goat breeding), (iii) the impacts of goat breeding practices on XT vegetation, and (iv) farmers' income from these three activities. The data concerning farm income from these three activities (SBA, WC production, and improved goat husbandry) are comparable because they concern the same periods (the 2010s and the 2020s), the same area (southwestern Madagascar with the same climate and vegetation), and the same farmers (farmers belonging to the Mahafaly and Tanalana ethnic groups).

### 3. Status of Malagasy XTs

Malagasy XTs are found in the southwest and the far south of Madagascar (Figures 1 and 2a). These areas have a semi-arid climate with mean annual rainfall that ranges from 300 to 800 mm [2,31], ≥9 dry months annually, and a water deficit > 400 mm [2,31]. XTs are restricted to an altitude below 400 m and cover approximately 17,000 km² (Figure 1; [32]). XT tree and shrub species show adaptations to drought, such as microphyllous (a reduction in the size of leaves) or aphyllous forms (a reduction in the number or a complete lack of leaves), spinescence (the development of thorns on branches instead of leaves), and pachycauly (swelling of the trunk, which allows water to be stored more efficiently) [2,33].

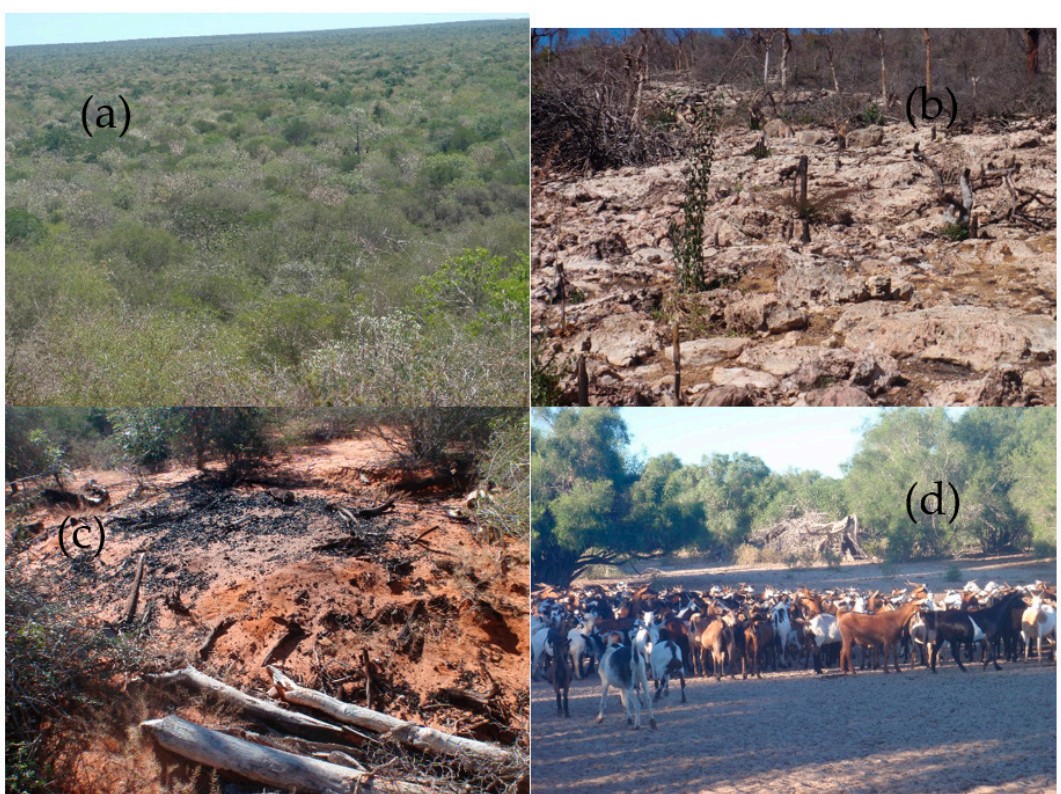

**Figure 2.** Xerophytic thicket (XT) landscape: (**a**) global view; (**b**) XTs after slash-and-burn agriculture; (**c**) a former wood charcoal kiln; (**d**) a goat herd.

Malagasy XTs lie essentially on two main soil types: calcisol (rocky calcareous soil) and lixisol (yellow/red sand soil), both of which are shallow and poor in organic matter and nitrogen [7,34]. In southwestern Madagascar, from the coastal zone to the inland area (from west to east), there are two main types of XT [2,35]:

- Short shrubby XT (Figure 2a), dominated by *Euphorbia* spp. (mainly *E. stenoclada* and *E. laro*) not exceeding 4 m in height, represents the most xeric formation. The only

*Didiereaceae* species occurring in this kind of XT are *Alluaudia comosa* and *Alluaudia fiherenensis*; and

- Tall XT with *Didierea madagascariensis*, *Adansonia fony*, and *Commiphora lamii* associated with an average height of 4–6 m and emergent trees reaching 8–10 m.

Malagasy XTs have a high percentage of plant endemism [36,37]: 48% of recorded genera and 95% of recorded species are endemic to Madagascar. For example, the *Didiereaceae* family has 11 species, of which 6 are endemic to Madagascar [33].

An expansion in cropland and further fragmentation of the remaining XTs are expected in the next three to four decades if no improvement is made in the livelihoods of households living in and around XTs, including food self-sufficiency [38]. Brinkmann et al. [3] demonstrated on the Mahafaly Plateau that SBA is mainly practised in remote areas far away from roads and markets. Randriamalala et al. [4] showed that SBA is more widely practised in the wetter, eastern part of the XT.

## 4. Xerophytic Thickets as Wooded Goat Rangeland

Malagasy XTs are browsed by goats (Figure 2d) [7,9]. Goat breeding in southwestern Madagascar is extensive and depends mainly on the availability of fodder in XTs [8,9,39,40]. Goats are bred for meat and milk production. The diet of goats consists mainly of the leaves of approximately 100 shrub species [9,39,40]. A partial list of these species is presented in Appendix A in terms of fitness for wood charcoal production and goat palatability estimated by the frequency of goat visits during one dry and one rainy season [40]. The diet of goats in the dry season consists of green leaves within their reach [39,40] or leaves, flowers, and fruits that have fallen to the ground from fodder shrubs. In the rainy season, the proportion of herbaceous species in the diet of goats increases but remains small (<35% of the daily feeding time; [9]). In addition to natural fodder, agricultural residues such as maize bran, cassava husks, and spoiled seeds and pods also contribute to the diet of goats [39]. They are available at the end of the rainy season, during the harvest period, and their consumption is limited in terms of quantity. The low use of feed supplements makes daily browsing times longer (10–13 h/day) [8,40]. The use of feed supplements may explain the relatively short browsing time (8–10 h/day) in other semi-arid areas (Senegal [41] and Oman [42]), as goats return to the pen earlier to receive feed supplements from crop residues. Goat herds in southwestern Madagascar may travel 7–15 km/day in the rangeland in search of food but do not stray more than 6 km from their night enclosure [8,40].

Studies on the impact of goat browsing on rangeland vegetation are scarce in Madagascar [7,13]. Ratovonamana et al. [13] showed that the combination of cattle and goat browsing of XTs in the littoral zone of southwestern Madagascar contributes to a reduction in the species richness. In contrast, Randriamalala et al. [7] showed that goat browsing alone has a slight effect on XT ecosystems. The species richness, regeneration rate, and leaf biomass (≤3 cm in diameter) of southwestern XTs were found to not vary significantly with goat browsing intensity in a comparison between a grazed site with a stocking rate of one head per hectare and a site too far away to be grazed that had not been cleared for SBA or for WC production [7]. This discrepancy may stem from the fact that the sites studied by Ratovonamana et al. [13] may have been affected by other practices, such as WC production and SBA. The western XTs in the littoral part of southwestern Madagascar are former charcoal production sites (charcoal production at these sites ceased in the early 2000s due to the depletion of wood resources) [39,43]. Thus, it is reasonable to assume that the effects of goat browsing on XTs are slight at the current stocking rate of one head per hectare.

## 5. Main Causes of XT Deforestation and Degradation

### 5.1. Slash-and-Burn Agriculture, the Main Cause of Deforestation in XTs

Post-SBA recovery in dense humid forests [44–47] and in dense dry forests [48–50] is fairly well documented in Madagascar. In contrast, studies on post-SBA recovery of XTs are scarce [35,51]. The species richness and floristic composition of post-SBA XTs vary

significantly between soil types: XTs on calcisol are richer in species than XTs on lixisol. The species composition and richness of overstory XTs (>1.3 m in height) have not changed since the farmers abandoned the plots. However, structural parameters such as total height, basal area, and mean diameter at breast height have increased significantly since plot abandonment, irrespective of the soil type. The slowness of the growth of shrub species found in post-cultivation XT plots may explain this lack of a change in the diversity and floristic composition through secondary succession stages [52]. For example, for nine XT species, it would take between 30 and 130 years to reach a diameter at breast height of 10 cm and over 100 years to reach a diameter of 30 cm [52].

*5.2. Wood Charcoal Production, a Major Source of XT Degradation*

All of the WC supplying the town of Toliara, the main city in southwestern Madagascar, comes from the XTs and the dense dry forests of southwestern Madagascar [11,14,53]. Approximately 43,000 t of WC are consumed annually by the town of Toliara [54], which is equivalent to an annual wood loss of over 340,000 t based on a carbonization yield of 0.125 [55]. The fuelwood market generally ensures the subsistence of poor households and is an important source of income during the dry season [43,56–58]. The semi-arid climate of southwestern Madagascar is characterized by a succession of drier and wetter periods, both of which last from 3 to 7 years [58]. Crop harvests are particularly poor during drier periods when WC production activities intensify [17]. All households produce WC over a period of 1 to 12 months, depending on their wealth [43,56]. Wood charcoal production is also a gap-filler during the agricultural off-season and a relatively rapid way to generate cash at short notice [57]. Wealthier households with more livestock (zebus and goats) engage in this activity for a shorter period (<3 months) and at the end of the dry season around October–November [43].

Wood charcoal production involves a significant loss of wood biomass, which is therefore less available in the vicinity of villages/settlements [7,10]. This biomass scarcity has led to the exploitation of tamarind trees for charcoal wood, despite their important cultural value [59]. Wood charcoal production is not a sustainable activity and would lead to the extinction of hardwood species suitable for making WC at current production sites in less than half a century [11]. Although there are numerous WC species in the XT (a partial list is given in Appendix A) [10,11], their extinction as a result of overexploitation of mature individuals may indeed occur. Changes in the main WC production sites around the town of Toliara confirm this lack of sustainability [54]. The search for sustainability in the exploitation of XTs, mainly for the purpose of WC commercialization, is difficult because of their low productivity with respect to woody biomass. Reducing the pressure on this vegetation type by planting fast-growing exotic species, such as *Eucalyptus* spp. and *Acacia* spp., at the wetter sites to the north and/or east of the XTs may be suggested as a way to preserve them [11,54,56]. Similarly, the planting of exotic multi-use species, such as *Ziziphus* spp. (fodder and charcoal), at sites close to the villages around the XTs may also be recommended to reduce the pressure on these thickets [11,39].

## 6. Goat Breeding: An Alternative to Wood Charcoal Production and Slash-and-Burn Agriculture in XTs

Goat breeding in southwestern Madagascar is extensive and suffers from low production rates due mainly to (i) long inter-gestation periods [60] and (ii) the high rate of mortality of kids reported by farmers [39,61]. Despite these limitations, goat breeding is an important source of income for farmers [17,43]. Indeed, goat sales constitute more than 50% of the annual household cash income in the commune of Soalara-Sud [43] and cover 15–29% of the net food expenditure on the entire Mahafaly Plateau (compared with less than 6% for food crops; [17]). However, the key question regarding improvements to goat breeding practices is: would this income incentive be sufficient to persuade the farmer to breed goats rather than to cultivate the land or to produce WC? Improved husbandry practices can indeed significantly increase the production of livestock and thus the income obtained from

it. However, this improvement must be simple and not involve many changes in current pastoral/breeding practices. The batch breeding technique satisfies these conditions. It consists of dividing the herd into the following batches: (i) young and lactating females; (ii) adults (reproductive animals); and (iii) castrated males and culled adults. The animals from the last batch and the surplus animals from the second batch are intended for sale. The application of this breeding method involves oestrus synchronization to obtain homogeneous batches of animals and to improve the reproductive performance of the goats. Flushing, which is known to have positive effects on small ruminant reproduction [62,63], is a simple technique used to achieve this oestrus synchronization. Andrianarisoa et al. [61] conducted farm trials in southwestern Madagascar on the control of goat reproduction using the flushing technique. The treatment consisted of feeding young females dry cassava at a rate of 500 g/day/individual for 45 days in June–July, a period of calving, and the availability of a small amount of fodder in the rangeland [61]. Cassava is available locally, and dry cassava can be purchased at local markets at a low cost. The aim is to change the birth period from the dry season to the short rainy season when shrubs have leaves.

Flushing significantly improves several parameters of goat reproduction [61], including the fertility rate (the ratio of the number of females giving birth to the number of bred females), the fecundity rate (the number of kids born in relation to the number of bred females), and the survival rate at one month from birth (the ratio of the number of kids alive at one month to the number of kids born alive). Thus, the use of this oestrus synchronization technique would enable 1.5 (3 parturitions/2 years) to 2 parturitions per year instead of 1 (goats in southwestern Madagascar give birth every 12.4 months on average; [61]) in addition to producing homogeneous batches of animals. Batch breeding expenses were related to the purchase of feed supplements for the dry period (dried cassava; USD 2.90/head/year) and deworming products (USD 0.15/head) to ensure the comparability of treated and control goat groups [61]. All of these products are locally available. The potential income from the sale of surplus produced animals increases as the number of bred females increases (Figure 3). The challenge is to manage both the increase in income from goat breeding and the number of young females, which is limited by the XT carrying capacity.

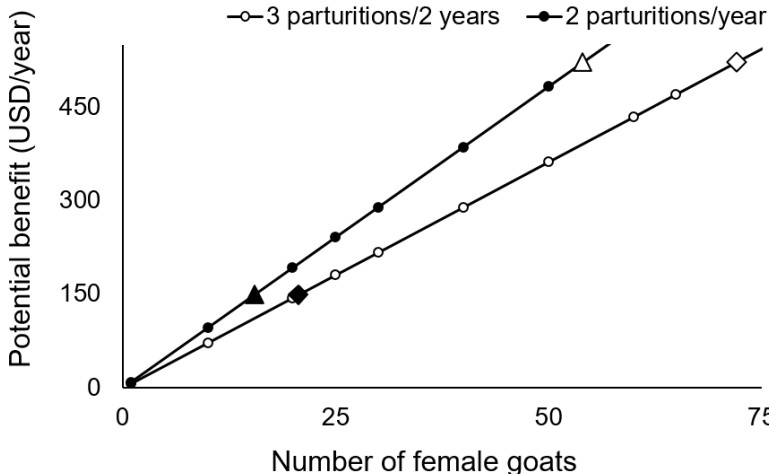

**Figure 3.** Potential benefit from the simultaneous application of the flushing and batch breeding techniques [61]. The black triangle and black diamond denote the minimum numbers of female goats required to replace the income from wood charcoal production (15 and 21 goats, respectively); the white triangle and white diamond denote the minimum numbers of female goats required to replace the income from wood charcoal production and slash-and-burn agriculture (54 and 72 goats, respectively).

The data on income from SBA are from Neudert et al. [64], who estimated the opportunity costs of conserving Malagasy XTs (Table 1). The income associated with SBA is equivalent to the estimated opportunity cost of giving up that activity. The data on income

from WC production are from Neudert et al. [64] and Masezamana et al. [56], who studied the value of the WC production chain in southwestern Madagascar (Table 1).

**Table 1.** Income from slash-and-burn agriculture [64,65].

| N | Parameters | Amount (USD) |
|---|---|---|
| I | Net income (USD/ha/year) [61] | 177.4 |
| II | Mean crop field area [62] | 2.1 |
| III | Profit from SBA (USD/ha/year) (=I × II) | 372.54 |

A simple linear regression from the data of Andrianarisoa et al. [61] (Figure 3) was used to estimate the minimum number of female goats necessary to obtain the calculated income from SBA and WC production according to data from [56,64,65] (Tables 1 and 2). The mean annual income from SBA and WC production together, as a net of different production costs, is about USD 521 (=372 + 149; Tables 1 and 2). Thus, between 15 (2 parturitions/year) and 21 (3 parturitions/2 years) female goats would have to be bred to produce the equivalent annual income to WC production. However, between 54 (2 parturitions/year) and 72 (3 parturitions/2 years) female goats would have to be bred to produce the equivalent annual income to that from SBA and WC production. This is twice or three times the average small ruminant herd size estimated by Neudert et al. [65] on the Mahafaly Plateau, which is 25 heads per farm. Thus, it is possible to increase the income obtained from goat breeding, which could replace the income from WC production and a large part of the income from SBA agriculture, but this would have to start with increasing the average herd size (from 25 to 54–72 head). These new increased herd sizes might lead to a doubling or tripling of the stocking rate in the XTs, which would entail ecological risks, and the increased income would be subject to socio-economic risks related to market and price instability, which are discussed below.

**Table 2.** Income from wood charcoal production [53,61].

| N | Parameters | Locations | |
|---|---|---|---|
| | | XT North of Onilahy River [a] | XT South of Onilahy River [b] |
| I | WC producer sample size (household) [56] | 120 | 113 |
| II | Annual duration of WC production (months) [56] | 8.85 | 6 |
| III | Monthly WC production (kg) [56] | 1400 | 759 |
| IV | Annual WC production (kg/year) (=II × III) | 12,390 | 4554 |
| V | Mean weight of a WC bag (kg) [64] | 23 | 23 |
| VI | WC bag number (=IV/V) | 538.70 | 198 |
| VII | Price/bag (USD) [64] | 0.4 | 0.4 |
| VIII | Annual income/household from WC production (USD) (=VI × VII) | 215.48 | 79.2 |
| IX | Average annual income from WC production (USD) (=[(VIII[a] × I[a]) + (VIII[b] × I[b])]/(I[a] + I[b]) | 149.37 | |

## 7. Recommendations for Sustainable Goat Breeding

### 7.1. Fodder Availability and Rangeland Management

The application of the batch breeding technique combined with flushing as an oestrus synchronization method would significantly increase the size of the goat herds in southwestern Madagascar in a short period of time (a doubling or tripling of the size in less

than 5 years). This situation may lead to overgrazing and the negative effects this could have on wooded rangeland, such as a reduction in plant cover [20–22] and a disruption in the regeneration of fodder species [23,24]. An assessment of the XT carrying capacity for goats—the maximum population size that it can support indefinitely [66]—is necessary. An accurate assessment of the carrying capacities of existing rangelands and simulations of the effects of the adoption of flushing and batch husbandry on goat herd size should be carried out. The negative effects of goat browsing on vegetation in semi-arid areas are mainly a consequence of an animal stocking rate that is too high [29]. The carrying capacity of a rangeland is a function of (i) its annual fodder biomass production, (ii) the daily amount of fodder ingested by the ruminants, and (iii) the annual browsing duration. This assessment of the carrying capacity of the XT, which is a dry wooded rangeland that has spiny vegetation rather than leaves during the long dry period, presents methodological challenges. Carrying capacity assessments have generally concerned herbaceous fodder in rangeland [67–70]. The contribution of shrubs to the availability of forage in wooded rangelands is recognized [71–74]. In contrast, the annual fodder biomass production of shrubs has often been assessed at experimental sites [75,76] and rarely in rangeland [77]. The annual increase in fodder biomass is part of the net primary productivity of the ecosystem and is more difficult to estimate for shrubby fodder species. It may be estimated by the ratio between the shrub fodder biomass and its age. This latter element can be equated to the number of annual growth rings [11,52].

There are two possible scenarios: (i) the XT carrying capacity is well above the higher stocking rate corresponding to an average herd size of 54–72 heads/farm; or (ii) the current stocking rate of 1 head/ha is close to the XT carrying capacity. In the first case, the improvement of goat husbandry proposed in this work is ecologically sustainable since the increase in the goat herd size would not correspond to a degradation of the XT (no significant effect on the natural regeneration of fodder species or on the availability of fodder biomass). In the second case, the proposed improvement of goat husbandry would lead to XT over-browsing and to degradation. In the latter case, it would be recommended to maintain the current stocking rate. In this case, goat breeding would remain an important livelihood component but would not constitute a sustainable alternative to the combination of WC production and SBA; it could only supplement the income from WC production.

In all cases, increasing the availability of fodder by planting fodder species would be recommended. However, this would only produce significant results in several years (>10 years) because of the aridity of the climate, which slows down the growth of shrubs [11,52] and also reduces the survival rate of seedlings [11,51]. Better knowledge of the ecology (distribution, growth, and limiting factors) and biology (natural regeneration and optimal multiplication techniques) of shrubby species in XTs is essential for successful planting of fodder species. Furthermore, planting at the beginning of the rainy season and regular monitoring of the seedlings would significantly increase the survival rate of the planted seedlings. Native fodder species adapted to the local arid climate should be prioritized to minimize the risk of invasion of XTs by exotic species. Indeed, native species could be planted in gaps and post-SBA regrowth areas, while exotic species such, as *Ziziphus* spp., could be planted in courtyards and around crop fields near villages.

### 7.2. Necessary Reorganization of the Goat Value Chain

An increase in the herd size, which is easy to achieve [61], offers an opportunity to increase the farm's income if the market is organized in such a way as to clearly identify a demand for goat meat and if the price of this product is stable. At the moment, the goat value chain is not properly organized. Purchases are made directly from the farmer or at the marketplace in each rural commune. Goat sales are made on the basis of pressing need and are not planned [17]. These factors make it difficult for goat traders to acquire a large number of animals in a short period of time. Another limitation is the difficulty of selling surplus animals as assets to be made profitable when goat herds are in fact perceived as savings that can be mobilized in case of need (e.g., illness, traditions related to marriage and

burial, a loss of crops [17,43]) and not as assets to be made profitable. To avoid these pitfalls, the goat value chain should be reorganized. First, goat farmers should form associations or cooperatives to provide them with technical (awareness of and training on how to breed in batches and herd and rangeland management) and organizational (financial and administrative management) support. Such activities depend on development policies, which are the responsibility of the State with the help of development NGOs. Then, they should promote the establishment of links to other stakeholders in the goat value chain (butchers, live animal traders, meat or live animal exporters, etc.). These other stakeholders could also be encouraged to assist the State and development NGOs in providing technical and organizational support to goat farmers in order to ensure the sustainability of goat meat or live animal supplies.

Goat farming in southwestern Madagascar can be sustainable and would be a factor in the development of this region if the sale of the surplus animals produced could be guaranteed. The conditions to ensure the sale of surplus animals produced can be satisfied in Madagascar because: (1) there is a demand for meat to partially replace zebu meat, which has become deficient [78]; (2) there is a demand for live animals for exports to the Saudi Arabian or Persian Gulf markets; (3) there is a demand for goat meat for Muslim countries instead of the more highly valued mutton for the end of Ramadan; (4) the price can be sustained as long as supply remains below demand at the regional level; and, in the case of the export sector, (5) livestock farming will remain in competition with the farming of dry grains or charcoal production for small-scale livestock farmers who do not have the capacity to increase the size of their herd sufficiently. It is advisable to develop local sales first, as they imply minimal restrictive standards and modest logistical means (in terms of homogeneity of animal batches, hygiene, animal welfare, labelling, the refrigeration chain, etc.). Then, exports to countries with requirements and standards that are more accessible to breeders in southwestern Madagascar can be targeted (Comoros and China).

## 8. Conclusions

The application of the batch breeding technique and flushing to improve goat reproductive parameters and stimulate oestrus synchronization may be a viable and sustainable alternative to WC production practices in the XT. On the other hand, an increase in the herd size and thus in the stocking rate is necessary in order for improved goat breeding to cover the combined income from WC production and SBA. The effects of such an increase in the stocking rate on the XT ecosystems would depend on their carrying capacity. In all cases, improvements in the reproductive parameters of goats make it possible to produce surplus animals that can be sold and thus increase the income of the farmers. However, it is essential to ensure the sale of surplus animals resulting from improvements in animal husbandry to avoid an excessive increase in the size of the herd. An improvement in the organization of the farmers is necessary, involving the formation of associations and/or cooperatives and training in herd and administrative management, and of the entire goat value chain associating producers with the other stakeholders in the sector in order to facilitate the sale of surplus animals produced. Finally, to ensure the sustainability of goat breeding, an assessment of goat–rangeland interactions with regard to carrying capacity is needed.

**Author Contributions:** Conceptualization and writing—original draft preparation, J.R.R.; writing—review and editing, J.R.R. and D.H. All authors have read and agreed to the published version of the manuscript.

**Funding:** This research received no external funding.

**Institutional Review Board Statement:** Not applicable.

**Informed Consent Statement:** Not applicable.

**Data Availability Statement:** Not applicable.

**Conflicts of Interest:** The authors declare no conflict of interest.

## Appendix A

**Table A1.** Main shrubby fodder species in the XT based on data from [7,40]. Palatability from the frequency of goat visits [42]: +++ Very high; ++ high; + mean; 0 Low.

| Species | Family | *Vernaculary Name* | Edible Part | Species Fit for Wood Charcoal Production | Palatability |
|---|---|---|---|---|---|
| *Acacia bellula* | *Fabaceae* | *Rohinala* | *Leaf* | *Yes* | 0 |
| *Albizia tulearensis* | *Fabaceae* | *Fengodiba* | *Leaf; fruit; flower* | *No* | + |
| *Allophylus decaryi* | *Sapindaceae* | *Sarivoamanga* | *Leaf* | *No* | 0 |
| *Allophylus dissectus* | *Sapindaceae* | *Sarivoamanga2* | *Leaf* | *No* | 0 |
| *Bauhinia grandidieri* | *Fabaceae* | *Seta* | *Leaf* | *Yes* | 0 |
| *Boscia longifolia* | *Capparaceae* | *Paky* | *leaf* | *No* | 0 |
| *Cayratia triternata* | *Vitaceae* | *Tehezandradraka* | *Leaf* | *No* | 0 |
| *Cedrelopsis grevei* | *Meliaceae* | *Katrafay dobo* | *Leaf* | *Yes* | 0 |
| *Celosia argentea* | *Amaranthaceae* | *Fofotse* | *Leaf* | *No* | + |
| *Chadsia flammea* | *Fabaceae* | *Sangan'akoholahy* | *Leaf* | *Yes* | +++ |
| *Chadsia* sp. | *Fabaceae* | *Rohidroitse* | *Leaf* | *Yes* | + |
| *Combretum grandidieri* | *Combretaceae* | *Kapikala* | *Leaf; fruit; flower* | *No* | 0 |
| *Commiphora lamii* | *Burseraceae* | *Ariarinaliotse* | Leaf | No | + |
| *Commiphora lasiodisca* | *Burseraceae* | *Vingovingo* | *Leaf* | *No* | + |
| *Commiphora marchandii* | *Burseraceae* | *Holidaro* | *Leaf* | *No* | ++ |
| *Commiphora simplicifolia* | *Burseraceae* | *Sengatse* | *Leaf* | *No* | ++ |
| *Croton* sp. | *Euphorbiaceae* | *Saribalahazo* | *Leaf* | *No* | 0 |
| *Cynanchum* spp. | *Apocynaceae* | *Try* | *Leaf* | *No* | 0 |
| *Dalbergia xerophila* | *Fabaceae* | *Hazobango* | *Leaf* | *Yes* | + |
| *Dichrostachys alluaudiana* | *Fabaceae* | *Havoa* | *leaf* | *Yes* | ++ |
| *Dicoma incana* | *Asteraceae* | *Vongo* | *Leaf* | *No* | +++ |
| *Dicraeopetalum mahafaliense* | *Fabaceae* | *Lovainafo* | *Leaf* | *Yes* | 0 |
| *Didierea madagascariensis* | *Didiereaceae* | *Sono* | *Leaf* | *Yes* | ++ |
| *Digoniopterys microphylla* | *Malpighiaceae* | *Kitoky* | *Leaf* | *No* | 0 |
| *Digoniopterys microphylla* | *Malpighiaceae* | *Vongo2* | *Leaf* | *No* | +++ |
| *Diospyros latispathulata* | *Ebenaceae* | *Sasimotse2* | *Leaf* | *Yes* | +++ |
| *Diospyros manampetsae* | *Ebenaceae* | *Fivikakanga* | *Leaf; fruit; flower* | *No* | 0 |
| *Euphorbia fiherenensis* | *Euphorbiaceae* | *Samantam-bazaha* | *Leaf* | *No* | 0 |
| *Euphorbia laro* | *Euphorbiaceae* | *Laro* | *Leaf* | *No* | 0 |
| *Euphorbia stenoclada* | *Euphorbiaceae* | *Samanta* | *Leaf* | *No* | 0 |
| *Givotia madagascariensis* | *Euphorbiaceae* | *Farahafatse* | *Fruit* | *No* | 0 |
| *Gouania mauritiana* | *Rhamnaceae* | *Timbatse* | *Leaf* | *No* | ++ |
| *Grewia leucophylla* | *Malvaceae* | *Sely* | *Leaf* | *Yes* | ++ |
| *Grewia tulearensis* | *Malvaceae* | *Hazofotse* | *Leaf* | *Yes* | + |
| *Gyrocarpus americanus* | *Hernandiaceae* | *Kapaipoty* | *Leaf; fruit; flower* | *No* | ++ |
| *Helinus ovatus* | *Rhamnaceae* | *Masokarany* | *Leaf* | *No* | + |
| *Hilsenbergia croatii* | *Boraginaceae* | *Lampana* | *Leaf* | *Yes* | 0 |
| *Hilsenbergia lyciacea* | *Boraginaceae* | *Mera* | *Leaf* | *Yes* | 0 |
| *Ipomoea bolusiana* | *Convolvulaceae* | *Moky* | *Leaf* | *No* | + |
| *Ipomoea* sp. | *Convolvulaceae* | *Velahy* | *Leaf* | *No* | 0 |
| *Kalanchoe beharensis* | *Crassulaceae* | *Mozy* | *Leaf* | *No* | + |
| *Karomia microphylla* | *Lamiaceae* | *Forim-bitike* | *Leaf* | *Yes* | ++ |
| *Maerua filiformis* | *Capparaceae* | *Somangy* | *Leaf* | *Yes* | + |
| *Mimosa delicatula* | *Fabaceae* | *Rohy* | *Leaf* | *Yes* | +++ |
| *Neobeguea leandriana* | *Meliaceae* | *Handinaombilahy* | *Leaf* | *Yes* | + |
| *Neobeguea mahafaliensis* | *Meliaceae* | *Handy* | *Leaf* | *Yes* | + |
| *Operculicarya decaryi* | *Anacardiaceae* | *Jabia* | *Leaf; fruit; flower* | *No* | 0 |

**Table A1.** *Cont.*

| Species | Family | *Vernacular Name* | Edible Part | Species Fit for Wood Charcoal Production | Palatability |
|---|---|---|---|---|---|
| *Operculicarya* sp. | *Anacardiaceae* | *Tarabia* | *Leaf* | *No* | +++ |
| *Paederia grandidieri* | *Rubiaceae* | *Tamboromantsy* | *Leaf* | *No* | 0 |
| *Poupartia silvatica* | *Anacardiaceae* | *Sakoa* | *Leaf; fruit; flower* | *No* | 0 |
| *Pristimera bojeri* | *Celastraceae* | *Timbatse2* | *Leaf* | *No* | ++ |
| *Rhigozum madagascariense* | *Bignoniaceae* | *Hazotaha* | *Leaf* | *Yes* | +++ |
| *Ruellia latisepala* | *Acanthaceae* | *Fobo-drano* | *Leaf* | *No* | 0 |
| *Ruellia* sp. | *Acanthaceae* | *Tseatsea* | *Leaf* | *No* | 0 |
| *Salvadora angustifolia* | *Salvadoraceae* | *Sasavy* | *Leaf; fruit; flower* | *Yes* | ++ |
| *Salvadora angustifolia* | *Salvadoraceae* | *Sasimotse* | *Leaf* | *No* | +++ |
| *Sclerocarya birrea* | *Anacardiaceae* | *Handimbohitse* | *Leaf* | *Yes* | + |
| *Secamone* sp. | *Apocynaceae* | *Hazomby* | *Leaf* | *Yes* | 0 |
| *Solanum bumeliaefolium* | *Solanaceae* | *Hazonosy* | *Leaf* | *Yes* | +++ |
| *Stereospermum euphorioides* | *Bignoniaceae* | *Fanony* | *Leaf* | *No* | ++ |
| *Stereospermum euphorioides* | *Bignoniaceae* | *Somontsoy* | *Leaf* | *Yes* | + |
| *Suregada boiviniana* | *Euphorbiaceae* | *Hazom-balala* | *Leaf* | *Yes* | 0 |
| *Talinella boiviniana* | *Talinaceae* | *Tsiroraha* | *Leaf* | *No* | +++ |
| *Talinella dauphinensis* | *Talinaceae* | *Koamanga* | *Leaf* | *No* | 0 |
| *Tarenna* sp. | *Rubiaceae* | *Lelaomby* | *Leaf; fruit; flower* | *Yes* | 0 |
| *Terminalia fatraea* | *Combretaceae* | *Talia* | *Leaf* | *Yes* | 0 |
| *Terminalia ulexoides* | *Combretaceae* | *Fatra* | *Leaf; fruit* | *Yes* | 0 |
| *Tetrapterocarpon geayi* | *Fabaceae* | *Fengoke* | *Leaf; fruit; flower* | *No* | + |
| *Tetrapterocarpon geayi* | *Fabaceae* | *Voaovy* | *Leaf* | *Yes* | ++ |
| *Thilachium seyrigii* | *Capparaceae* | *Pakamboa* | *Leaf* | *Yes* | 0 |
| *Vaughania interrupta* | *Fabaceae* | *Falimaharay* | *Leaf* | *Yes* | + |
| *Xerosicyos danguyi* | *Cucurbitaceae* | *Tapisaky* | *Leaf* | *No* | 0 |
| *Ziziphus spina-christi* | *Rhamnaceae* | *Tsinefo* | *Leaf; fruit* | *Yes* | +++ |

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
