# Peer review of "Goat Breeding: A Possible Sustainable Way to Manage Xerophytic Thickets in Southwestern Madagascar"

_land, doi:10.3390/land11030405_

Round 1

Reviewer 1 Report

Manuscript ID: land-1600549

Goat breeding, a possible sustainable way to manage xerophytic thickets in southwestern Madagascar

Review

I would like to emphasize that this paper is a kind of "regional meta review", whose objective is to present a series of research results on the management of XT in southwestern Madagascar, to put the results in an international context and to provide proposals in order to improve the management. The character of a "meta review" is underlined by the fact that within the bibliography, about 16 references are from the authors of this manuscript and play a central role in the paper under review.

It means that the review below will not follow the classic pattern of this kind of exercise. I will focus less on methodology and more on the conclusions and proposals that are drawn from the "meta review".

To answer a question often asked to reviewers, the manuscript is not original in its "meta review" part. The originality of the paper lies in the conclusions and proposals to improve the management of a natural resource, e.g. the XT of southwestern Madagascar, for the benefit of the local population, based on their previous research as well as on international references.

For the clarity of my following remarks, I indicate that I know the authors of this paper and that I have already reviewed several of their papers submitted to different journals or financing institutions.

The general problem as well as the objective of this paper are very correctly presented in the introduction (chapter 1).

As already told above, given the peculiarity of the manuscript it is hardly possible to expect a detailed presentation of the methods implemented (chapter 2). In fact, this chapter is very concise. Some methodological indications given later, within the different chapters, complete the methodology. For my part, I have no criticism concerning this way of doing things.

Chapter 3 is ok on the whole, but it happens that certain assertions are not sufficiently explained. The following sentence for example "SBA is not the most profitable use of XT...", a statement which is not confirmed by the data in tab. 1 and 2.

Chapter 4 is ok. However, to give more consistency to the discussion of the goat browsing impact, chapter 4 and chapter 5.3 could be combined.

Chapter 5. From now on, the results and the discussion are mainly focused on SW Madagascar. Chapter 5.1 and 5.2 provide an interesting overview of the impact of SBA and WC on the XT. Chapter 5.3 is not strong enough and does not contain enough research results (incl. international) on the important question of the impact of goat browsing. I feel for example that the database is very (too) weak to assume that “… the effects of goat browsing on XT are slight …”.

As proposed above the combination of chapters 4 and 5.3 could improve the relevance of this part of the overview to some extent.

Chapter 6. This chapter explains with focus on the malagasy XT under what conditions (from the authors point of view) goat breeding could replace WC and/or SBA. The question mark in the title of the chapter could therefore be deleted. That said, the chapter is interesting, informative, and documented in a relevant way.

Chapter 7. In chapter 7.1, the question of fodder availability is central as it is in the whole paper. Recommendations are presented clearly and relevantly. However, when recommendations run into obstacles, the discussion tends to avoid the discussion on implementation problems. For instance the proposal of planting fodder species faces two problems: i) the slow growth due to the climate (point mentioned but not really discussed) and ii) the conservation of biodiversity, an item never addressed neither here nor above (lines 171 ff.). Such well-known implementation problems are crucial and should be discussed.

For various reasons, chapter 7.2, which most clearly addresses the question of the interface between human being and the natural resource, is quite problematic. Here are my points of disagreement. 1) The statement that "The increase in herd size should not be difficult to achieve..." is in clear contradiction to the last part of chapter 7.1. 2) Proposals for the reorganization of the value chain overuse formulations like "should be", which do not commit anyone and will not really move things forward if not properly discussed in terms of implementation. 3) The option of selling cattle or meat locally or abroad is not discussed in terms of difficulties and uncertainties of goat breeding in this region of Madagascar. Which recommendations are relevant? What is it possible to envisage, in a realistic way, therefore with chances of success? In this context, it is important to also think in terms of development policy. 4) The last paragraph of 7.2 is questionable because the recommendations are not really supported by experimental results or research findings. Statements such as "...this kind of goat breeding would also contribute to the conservation of XT and their biodiversity..." should be discussed more carefully with a bit of self-criticism.

Chapter 8 has the merit of being brief. I will however point out the last sentence and in particular that "... to ensure the sustainability of goat breeding, further knowledge ... is needed.". There is something classic about such a -justified - sentence. The problem is that it contradicts many of the given recommendations. This kind of sentence is too general. Research needs should be more clearly linked to problems raised and research items could be proposed in a slightly more precise way.

Appendix 1. This appendix is quite long. It has the merit of focusing on some of the authors' previous research work. However, it has the disadvantage of not bringing much to this paper. It would be enough to mention useful elements if necessary, for example at lines 115 ff.

Proposed corrections (not exhaustive)

Line 28-29: add SBA to the keywords

Line 33: Euphorbiaceae

Line 49: what is the meaning of “At international level…” in the present context? Maybe the sentence should begin like this: “At international level, many research papers show that …”

Line 102: Didiereaceae

Line 139: “… since plot abandonment by the farmers.”

Line 240- 241: Tab. 3 does not exist

Line 241: “… net annual income …” ok for SBA (cf. table 1), not clear in the case of WC (cf. table 2; please indicate that the given amounts are net)

Line 468: Amount instead of Montant

Line 470: Table 2 or 3?

Table 2: unit for WC producer sample size is missing

Appendix 1: Hernandiaceae

My conclusion is that this paper cannot be accepted as it stands and requires a major revision. According to my review the revision should focus on chapter 3, 4, 5, 7 and 8.

Author Response

RV1

Goat breeding, a possible sustainable way to manage xerophytic thickets in southwestern Madagascar

Review

I would like to emphasize that this paper is a kind of "regional meta review", whose objective is to present a series of research results on the management of XT in southwestern Madagascar, to put the results in an international context and to provide proposals in order to improve the management. The character of a "meta review" is underlined by the fact that within the bibliography, about 16 references are from the authors of this manuscript and play a central role in the paper under review.

It means that the review below will not follow the classic pattern of this kind of exercise. I will focus less on methodology and more on the conclusions and proposals that are drawn from the "meta review".

To answer a question often asked to reviewers, the manuscript is not original in its "meta review" part. The originality of the paper lies in the conclusions and proposals to improve the management of a natural resource, e.g. the XT of southwestern Madagascar, for the benefit of the local population, based on their previous research as well as on international references.

For the clarity of my following remarks, I indicate that I know the authors of this paper and that I have already reviewed several of their papers submitted to different journals or financing institutions.

The general problem as well as the objective of this paper are very correctly presented in the introduction (chapter 1).

As already told above, given the peculiarity of the manuscript it is hardly possible to expect a detailed presentation of the methods implemented (chapter 2). In fact, this chapter is very concise. Some methodological indications given later, within the different chapters, complete the methodology. For my part, I have no criticism concerning this way of doing things.

Chapter 3 is ok on the whole, but it happens that certain assertions are not sufficiently explained. The following sentence for example "SBA is not the most profitable use of XT...", a statement which is not confirmed by the data in tab. 1 and 2.

Authors : The sentence “SBA is not the most profitable use of XT...", with an unclear idea has been deleted.

Chapter 4 is ok. However, to give more consistency to the discussion of the goat browsing impact, chapter 4 and chapter 5.3 could be combined.

Authors : The chapters 4 and 5.3 have been combined (cf. L130-144).

Chapter 5. From now on, the results and the discussion are mainly focused on SW Madagascar. Chapter 5.1 and 5.2 provide an interesting overview of the impact of SBA and WC on the XT. Chapter 5.3 is not strong enough and does not contain enough research results (incl. international) on the important question of the impact of goat browsing. I feel for example that the database is very (too) weak to assume that “… the effects of goat browsing on XT are slight …”.

As proposed above the combination of chapters 4 and 5.3 could improve the relevance of this part of the overview to some extent.

 Authors : The effects of goat browsing on vegetation were explained in the introduction (L 48-57). The two local references (Randriamalala et al. [7] and Ratovonamana et al. [13]) were compared and discussed with international references.

Chapter 6. This chapter explains with focus on the malagasy XT under what conditions (from the authors point of view) goat breeding could replace WC and/or SBA. The question mark in the title of the chapter could therefore be deleted. That said, the chapter is interesting, informative, and documented in a relevant way.

Authors : Done (question mark deleted).

Chapter 7. In chapter 7.1, the question of fodder availability is central as it is in the whole paper. Recommendations are presented clearly and relevantly. However, when recommendations run into obstacles, the discussion tends to avoid the discussion on implementation problems. For instance the proposal of planting fodder species faces two problems: i) the slow growth due to the climate (point mentioned but not really discussed) and ii) the conservation of biodiversity, an item never addressed neither here nor above (lines 171 ff.). Such well-known implementation problems are crucial and should be discussed.

Authors : Additional recommendations were provided to improve the success of plantations and to limit the risk of invasion by alien species. (cf. L 294-302). In addition, the paragraph on "rainfall variability" has been deleted as we feel that this issue related to the equilibrium character or not of the goat/rangeland interaction deserves to be addressed in a separate article.

For various reasons, chapter 7.2, which most clearly addresses the question of the interface between human being and the natural resource, is quite problematic.

Here are my points of disagreement.

  • The statement that "The increase in herd size should not be difficult to achieve..." is in clear contradiction to the last part of chapter 7.1.

Authors: The reasons for the ease of increasing herd size were explained at the beginning of section 7.1 (L257-259): combination of batch breeding and flushing according to the results of [63]. The problem is that it is not known whether there is enough fodder to feed the surplus animals in XT. And this point is discussed at the end of section 7.1 (carrying capacity and plantation of fodder species)

  • Proposals for the reorganization of the value chain overuse formulations like "should be", which do not commit anyone and will not really move things forward if not properly discussed in terms of implementation.

Authors: Those responsible for the recommended activities are named in the text (State, NGO and other stakeholders in the goat sector). Changes have been made in the wording for more clarity (cf. L315-323).

3) The option of selling cattle or meat locally or abroad is not discussed in terms of difficulties and uncertainties of goat breeding in this region of Madagascar. Which recommendations are relevant? What is it possible to envisage, in a realistic way, therefore with chances of success? In this context, it is important to also think in terms of development policy.

Authors: The difficulties of selling are already discussed in L307-315. Local sales are to be prioritised and exports are options for later when farmers are more professional and the goat value chain more stable (see add in L336-339).

  • The last paragraph of 7.2 is questionable because the recommendations are not really supported by experimental results or research findings. Statements such as "...this kind of goat breeding would also contribute to the conservation of XT and their biodiversity..." should be discussed more carefully with a bit of self-criticism.

Authors: This paragraph has been deleted as the ideas they convey are repeated in the conclusion and other sections.

Chapter 8 has the merit of being brief. I will however point out the last sentence and in particular that "... to ensure the sustainability of goat breeding, further knowledge ... is needed.". There is something classic about such a -justified - sentence. The problem is that it contradicts many of the given recommendations. This kind of sentence is too general. Research needs should be more clearly linked to problems raised and research items could be proposed in a slightly more precise way.

Authors: “further knowledge” has been changed by “assessment” and references to "rainfall variability" have been removed (see comments above) (cf. L365)

Appendix 1. This appendix is quite long. It has the merit of focusing on some of the authors' previous research work. However, it has the disadvantage of not bringing much to this paper. It would be enough to mention useful elements if necessary, for example at lines 115 ff.

Authors: This appendix has been kept because it illustrates the abundance of fodder species and also gives an overview of WC species (see addition in L180-181).

Proposed corrections (not exhaustive)

Line 28-29: add SBA to the keywords

Authors : Done.

Line 33: Euphorbiaceae

Authors : Done.

Line 49: what is the meaning of “At international level…” in the present context? Maybe the sentence should begin like this: “At international level, many research papers show that …”

Authors : Done. The beginning of the sentence has been changed as proposed.

Line 102: Didiereaceae

Authors : Done.

Line 139: “… since plot abandonment by the farmers.”

Authors : Done.

Line 240- 241: Tab. 3 does not exist

Authors : You’re right. These are Tab 1 and 2 and not Tab 2 and 3.

Line 241: “… net annual income …” ok for SBA (cf. table 1), not clear in the case of WC (cf. table 2; please indicate that the given amounts are net)

Authors : Done.  We reformulate this sentence about the unclear “net income”: “The mean annual income from SBA and WC production together, both of them net of different production costs, is about USD 521….” (cf. L241-242)

Line 468: Amount instead of Montant

Authors : Done. Sorry for that.

Line 470: Table 2 or 3?

Authors : The same mistake…this is : Tab. 2.

Table 2: unit for WC producer sample size is missing

Authors : The unit for WC producer sample size is the number of households that produce WC.

Appendix 1: Hernandiaceae

Authors : Done.

My conclusion is that this paper cannot be accepted as it stands and requires a major revision. According to my review the revision should focus on chapter 3, 4, 5, 7 and 8.

Reviewer 2 Report

The article can be accepted at its current form.

Author Response

Thank you for the second reviewer for accepting the manuscript at its current form.

Round 2

Reviewer 1 Report

The authors have reworked their text to a large extent according to the observations and suggestions of the reviewer. The paper is now, in my opinion, ready for publication.

This manuscript is a resubmission of an earlier submission. The following is a list of the peer review reports and author responses from that submission.

Round 1

Reviewer 1 Report

It’s import to deter deforestation and degradation, and find a way to sustain the development of the far south and south-west of Madagascar. But I am afraid that this article cann’t be published in Land, due to lack of logic in writing, lack of quantitative data and analysis, poor English writing, confused literature insertion and so on.

Detailed income and expenditure of goat breeding wasn’t showed and discussed.

Abbreviation should be given at the first time appearing in abstract, text, and figure. Then the full name should be replaced by abbreviation at all the other places.

Line 152-154: The charcoal product depleted 20 years ago, so there was no meanings to assess goat breeding alternative to wood charcoal production.

Author Response

Reviewer 1

It’s import to deter deforestation and degradation, and find a way to sustain the development of the far south and south-west of Madagascar. But I am afraid that this article cann’t be published in Land, due to lack of logic in writing, lack of quantitative data and analysis, poor English writing, confused literature insertion and so on.

Authors: The manuscript has been restructured. In particular, a Method section explaining how the data were obtained and their nature has been added.

This is a literature review on the sustainability of XT management in south-western Madagascar. The added value of this paper is that it demonstrates the possible sustainability of improved goat husbandry in XT using existing/second-hand data. Similar exploitation of the data has not yet been done elsewhere and is justified by the fact that these second-hand data concern the same populations, living in the same bioclimatic area (south-western Madagascar) during the same periods (decades 2010-2020) and having the same lifestyles.

The English of this new version has been improved and literature citation checked and corrected.

Detailed income and expenditure of goat breeding wasn’t showed and discussed.

Authors: Details on improved goat husbandry have been added to the text (cf. line 233-236). The goal of the paper was not goat breeding alone, where income and expenditure related to goat breeding should have been detailed, but a comparison of sustainability between goat breeding, slash-and-burn agriculture and wood charcoal production, all three activities based on xerophytic thickets resources.

Abbreviation should be given at the first time appearing in abstract, text, and figure. Then the full name should be replaced by abbreviation at all the other places.

Authors: Done. The XT abbreviation has been substituted for for xerophytic thicket in the Abstract, text and figure after the first mention.

Line 152-154: The charcoal product depleted 20 years ago, so there was no meanings to assess goat breeding alternative to wood charcoal production.

Authors: This depletion of wood resources concerns the western part of the XT, the part near the sea (the most westerly) but not all the XT. Furthermore, the charcoal production sites are located around the villages and do not concern the whole XT. But these production sites would expand to larger areas if no effective alternative to charcoal production is found. The sentence in line 196-197 has been clarified.

Reviewer 2 Report

Comments from the reviewer

Lines

Comments

Line number 14

Browsed or pastured?

17-20

Is this a review or an article?

Line 31

Check the species spelling

Line 32

You don’t start the sentence with abbreviation. Do that throughout the manuscript

Line 50-53

You need to use one word. either browse or grazing. goats are normally browsing animals and this can suit well within the scope of this manuscript.

Line 55

Maybe it can come before ‘However……” sentence

There is a need to introduce the another subheadings focusing on

1. The state of XT in Madagascar.

2. The threat these thickets might have on ecosystem. Because this thicket (dense or encroachment) can contribute little no ecological niche to the economy. It can also be a habitat for predators which may even pose threat to our livestock.

3. The feeding value of XT in Madagascar

NB: There is a need to put the pictures of the type of xerophytic thickets species found to improve the readership of the manuscript.

NB: Examples of these species in a table form. containing some of these species or varieties, feeding value can be an ideal. All species that are forming part of these thickets with their usefulness to livestock (this will assist the readers on which ones are poisonous and which one are edible. which part of plant is mostly preferred).

NB The feeding value of most dominant xerophytic species. (looking at proximate analysis review (CP Minerals together with anti-nutritional factors), in a table format.

In the introduction. The authors did not mention the names of goat breeds available in the country. Indigenous or exotic?

Line 82-84

Observed where?

Line 88-90

Maybe there is a need to highlight the reasons for the lack of change in diversity.

100-101

Maybe there is a need to indicate how long does it take for one climate to take place. Because it sounds like there was a balance between the two climatic conditions.

145

One would ask why are we introducing goats as an alternative when the experiment did not show any significant effect

164

Is the method doable or feasible enough to be applied by farmers especially small and subsistence farmers, because even this manuscript is not indicating which group of farmers are supposed to benefit from this info.

This manuscript did not state whether all these plant species are available in both communal and commercial livestock areas.

204-208

There is a contradiction in these sentences. one would assume that where there is lack of education, livestock management practices such flushing, oestrus synchronisation and can be difficult to do.

217-220

Reference is required here

227

Actors or stakeholders. I think stakeholders is the most appropriate.

269

What is FX?

The manuscript can be improved if the authors can consider all these additional points I suggested, the authors can be given extra days to improve the manuscript.

Decision: MAJOR REVISION, provided they can do all the suggested comments.

Author Response

Reviewer 2

Line number 14          Browsed or pastured?

Authors: browsed is the correct term, inserted in the whole text.

17-20   Is this a review or an article?

Authors: It is a review. The type of paper has been changed.

In fact, the authors combine their published results with other published references to complete their demonstration about the sustainability of improved goat breeding on xerophytic thickets. This demonstration is new and original, not yet published. The type of paper is a review finalized by the demonstration on the sustainability of improved goat breeding.

Line 31 Check the species spelling

Authors: Done. Latin name of species in italic.

Line 32 You don’t start the sentence with abbreviation. Do that throughout the manuscript

Authors: Done

Line 50-53       You need to use one word. either browse or grazing. goats are normally browsing animals and this can suit well within the scope of this manuscript.

Authors: Done. Grazing has been changed to browsing.

Line 55 Maybe it can come before ‘However……” sentence

There is a need to introduce the another subheadings focusing on

  1. The state of XT in Madagascar.

Authors: Subhead on “status of Malagasy  XT” has been added.

  1. The threat these thickets might have on ecosystem. Because this thicket (dense or encroachment) can contribute little no ecological niche to the economy. It can also be a habitat for predators which may even pose threat to our livestock.

Authors: We did not add this new section because it is these XTs that are themselves threatened by human activities and should be preserved because of their biodiversity, future recognition as world heritage of humanity  and their usefulness to local populations (sources of food, wood, wooded rangeland for ruminants, medicinal plants, etc.).

  1. The feeding value of XT in Madagascar

Authors: A new section (=4. Xerophytic thickets as goat rangeland) has been added. Data on the feeding values of Malagasy XT fodder species are scarce and most XT fodder species have not yet been analysed. The same is true for toxicities and other chemical properties of XT fodder species. The interest of goat breeding in preserving these XT ecosystems may prompt such research in the future. Therefore, the main XT fodder species have been provided in the text (cf. new Tab. 1)

NB: There is a need to put the pictures of the type of xerophytic thickets species found to improve the readership of the manuscript.

Authors: Two photo series have been  inserted in the manuscript (figures 2 and 3).

NB: Examples of these species in a table form. containing some of these species or varieties, feeding value can be an ideal. All species that are forming part of these thickets with their usefulness to livestock (this will assist the readers on which ones are poisonous and which one are edible. which part of plant is mostly preferred).

Authors: Done, a table showing shrubby fodder species in XT has been added (new Table 1).

NB The feeding value of most dominant xerophytic species. (looking at proximate analysis review (CP Minerals together with anti-nutritional factors), in a table format.

Authors: Data on the feeding value of XT species are almost non-existent at the moment.

In the introduction. The authors did not mention the names of goat breeds available in the country. Indigenous or exotic?

Authors: Local goat Capra hircus has been added  in the Introduction (Line 13 and 44)

Line 82-84       Observed where?

Authors: Observed in the XT. This sentence has been rewritten (line 142-143).

Line 88-90       Maybe there is a need to highlight the reasons for the lack of change in diversity.

Authors: The explanation is given in the text: slowness of the growth of shrub species found in post-cultivation XT plots (cf. line 147-150).

100-101          Maybe there is a need to indicate how long does it take for one climate to take place. Because it sounds like there was a balance between the two climatic conditions.

Authors: Both drier and wetter climatic periods may last 3 to 7 yr. The sentence has been amended as follows:

“The semi-arid climate of south-western Madagascar is characterised by a succession of drier and wetter periods, both of them lasting from 3 to 7 years” (Line 161)

145      One would ask why are we introducing goats as an alternative when the experiment did not show any significant effect

Authors: Goat breeding is a traditional activity of households living in and around XT (cf. Introduction). International literature states that goat breeding contributes to degradation of vegetation in arid and semi-arid area and that it should be limited. The results in the XTs seem to show that goat browsing has little effect on the XT ecosystem. Therefore, goat breeding can be considered as an alternative to other activities that degrade XT (SBA and WC production).

164      Is the method doable or feasible enough to be applied by farmers especially small and subsistence farmers, because even this manuscript is not indicating which group of farmers are supposed to benefit from this info.

This manuscript did not state whether all these plant species are available in both communal and commercial livestock areas.

Authors: The XT is browsed only by small goat farmers, who are not yet involved in commercial breeding. To be feasible enough to be applied to farmers of this kind, the proposed improvement does not involve drastic changes in the current husbandry. For instance, to obtain oestrus synchronisation, a simple technique such as separating males is not feasible with the current livestock management practices. Flushing may be more practicable, because it uses local and cheaper dry cassava to stimulate female oestrus. The goats' nutrition will still mainly depend on xerophytic thickets, and proposed changes are limited to flushing, a one-off intervention at the beginning of the dry season period.

204-208          There is a contradiction in these sentences. one would assume that where there is lack of education, livestock management practices such flushing, oestrus synchronisation and can be difficult to do.

Authors: The explanation has been given in the previous answer to the reviewer.

217-220          Reference is required here

Authors: Authorization documents exist, delivered by the Ministry of animal breeding, to authorize the exportation to Comoros islands or to collect live goats. But these documents are private, including names of companies and numbers of the transaction, which we are reluctant to make public. What is your recommendation on this issue?

227      Actors or stakeholders. I think stakeholders is the most appropriate.

Authors: Done, stakeholders is the most appropriate.

269      What is FX?

Authors: You are right, FX is in French: “fourré xérophile”. The correct term is XT.

Round 2

Reviewer 1 Report

Comments and suggestions are showed in the pdf below.

Reviewer 2 Report

The manuscript is unacceptable at this stage 
